# Spermidine Enhances Mitochondrial Bioenergetics in Young and Aged Human-Induced Pluripotent Stem Cell-Derived Neurons

**DOI:** 10.3390/antiox13121482

**Published:** 2024-12-04

**Authors:** Leonora Szabo, Imane Lejri, Amandine Grimm, Anne Eckert

**Affiliations:** 1Research Cluster Molecular and Cognitive Neurosciences, University of Basel, 4002 Basel, Switzerland; 2Neurobiology Lab for Brain Aging and Mental Health, University Psychiatric Clinics Basel, 4002 Basel, Switzerland; 3Department of Biomedicine, University of Basel, 4055 Basel, Switzerland

**Keywords:** spermidine, aging, induced pluripotent stem cell-derived neurons, mitochondria, bioenergetics, oxidative stress

## Abstract

The accumulation of damaged mitochondria has long been considered a hallmark of the aging process. Among various factors, age-related mitochondrial alterations comprise bioenergetic impairments and disturbances in reactive oxygen species (ROS) control, thereby negatively affecting mitochondrial performance and ultimately accelerating aging. Previous studies have revealed that polyamine spermidine appears to exert health-protective and lifespan-promoting effects. Notably, recent findings have also described a spermidine-induced improvement in age-associated mitochondrial dysfunction, but the beneficial effects of spermidine on aged mitochondria have not been entirely examined yet. Here, we show that spermidine positively regulates several parameters related to mitochondrial bioenergetics and mitochondrial redox homeostasis in young and aged human-induced pluripotent stem cell-derived neurons. We report that spermidine treatment increases adenosine triphosphate production and mitochondrial membrane potential, which is accompanied by an attenuation in mitochondrial ROS levels in both age groups. Furthermore, we demonstrate a spermidine-mediated amelioration in mitochondrial respiration in both young and aged neurons. Overall, our findings suggest that nutritional spermidine supplementation might represent an attractive therapeutic approach to enhance mitochondrial function, consequently decelerating aging.

## 1. Introduction

For decades, global life expectancy has steadily increased. However, the exact reasons and mechanisms behind aging and how to support healthy aging are not entirely understood. In general, aging defines a gradual and multifactorial deterioration of cellular functions due to accumulated damage [1,2]. Notably, for many prevalent neurodegenerative diseases, such as Alzheimer’s disease or Parkinson’s disease, aging remains one of the leading risk factors [3]. Despite the common concept that the aging process involves many factors, typical age-dependent characteristics include genomic instability, telomere attrition, epigenetic alterations, loss of proteostasis, cellular senescence, and mitochondrial dysfunction [4].

Adult neurons are considered post-mitotic cells with a high oxygen demand to maintain neuronal function, integrity, and survival [5]. Therefore, it is not surprising that neurons are more vulnerable to disturbances in energy metabolism and thus more susceptible to accumulating defective mitochondria during aging [5,6]. Accordingly, healthy mitochondria are essential for the regulation of cellular energy generation in the form of adenosine triphosphate (ATP) through the process of oxidative phosphorylation (OXPHOS). In addition, changes in the bioenergetic status of mitochondria are closely linked to adaptations in mitochondrial morphology [7]. Mitochondrial dynamics, including fusion and fission, are vital in neurons to maintain functional mitochondria [8], with defects in these processes associated with neurodegeneration [9]. Furthermore, the PTEN-induced kinase 1 (PINK1)/Parkin pathway plays a key role in regulating mitochondrial quality control by selectively removing damaged mitochondria via mitophagy, thereby protecting neurons from oxidative damage and sustaining cellular homeostasis [10]. Equally fundamental, these organelles contribute to an array of other cellular functions, such as cell growth and differentiation, apoptotic pathways, intracellular calcium homeostasis, reduction-oxidation signaling, innate immunity, steroid biosynthesis, and synaptic plasticity [11,12]. Nevertheless, they also account for one of the major reactive oxygen species (ROS) sources, which, when formed in excess, can pose serious harm to neurons, eventually resulting in oxidative stress [13]. For this reason, mounting evidence indicates an association between the decline in mitochondrial function and neuronal aging [14]. In particular, a reduced oxidative capacity, diminished ATP generation, augmented ROS production, a weakened antioxidant defense system, and alterations in mitochondrial quality control are presumed to impact aging negatively [15]. Correspondingly, mitochondrial impairments have been recurrently appointed as potential drivers of aging [16]. Nonetheless, whether malfunctioning mitochondria cause aging or aging exacerbates mitochondrial dysfunction remains debatable.

Over the past years, various anti-aging interventions have emerged as promising strategies to extend a healthy lifespan or delay aging. Especially, the naturally occurring compound spermidine (SPD), belonging to the family of polyamines, was shown to exert cardioprotective, neuroprotective, and longevity-promoting effects [17]. More concretely, previous studies reported that dietary SPD supplementation decelerated aging in different model organisms [18,19,20,21] and decreased mortality rates in humans [22]. Intriguingly, recent findings described, in addition, an SPD-mediated amelioration of age-related alterations in mitochondrial function [18,19,23,24]. However, the protective effects of SPD on aged mitochondria have not been entirely investigated yet.

Hence, in the present study, we examined the impact of SPD on several parameters linked to mitochondrial bioenergetics and mitochondrial redox homeostasis in young and aged human-induced pluripotent stem cell (iPSC)-derived neurons. We demonstrate that SPD treatment augments ATP production and mitochondrial membrane potential (MMP), while simultaneously alleviating total mitochondrial ROS and specific superoxide anion radical (O_2_^•−^) levels in both age groups. Moreover, we report an SPD-induced improvement in mitochondrial respiration in both young and aged neurons.

## 2. Materials and Methods

### 2.1. Chemicals and Reagents

Dihydrorhodamine 123 (DHR), dimethyl sulfoxide (DMSO) Hybri-Max, Hanks’ Balanced Salt Solution (HBSS), spermidine BioUltra, and tetramethylrhodamine methyl ester perchlorate (TMRM) were purchased from Sigma-Aldrich (St. Louis, MO, USA). Cellartis DEF-CS 500 Culture System, Cellartis DEF-CS 500 COAT-1, RHB-A, RHB-BASAL, and NDiff N2 were obtained from Takara Bio (Kusatsu, Shiga, Japan). Advanced DMEM/F-12, B-27 Supplement (50X), distilled water, Geltrex LDEV-Free hESC-Qualified Reduced Growth Factor Basement Membrane Matrix, GlutaMax, Neurobasal Medium minus phenol red, PSC Neural Induction Medium, StemPro Accutase Cell Dissociation Reagent, and TrypLE™ Select Enzyme were all acquired from Gibco (Waltham, MA, USA). Seahorse XFp Cell Mito Stress Test Kit, Seahorse XF Calibrant Solution, Seahorse XF DMEM Assay Medium, pH 7.4, glucose, pyruvate, and glutamine were from Agilent Technologies (Santa Clara, CA, USA). MitoSOX™ Red Mitochondrial Superoxide Indicator and CellTracker Blue CMAC Dye were purchased from Invitrogen (Waltham, MA, USA). Recombinant Human EGF, Recombinant Human FGF-basic, and Recombinant Human/Murine/Rat BDNF were obtained from Peprotech (Cranbury, NJ, USA). ATPlite 1step Luminescence Assay was acquired from Perkin Elmer (Waltham, MA, USA), Phosphate-Buffered Saline (PBS) from Dominique DUTSCHER SAS (Bernolsheim, France), and Y-27632 (ROCK Inhibitor) from Selleck Chemicals (Houston, TX, USA).

### 2.2. Cultivation of Human iPSCs

Human iPSCs were either purchased from Takara or kindly provided by Dr. Zameel Cader (University of Oxford) and the Stem cells for biological assays of novel drugs and predictive toxicology (StemBANCC) consortium (Table 1). Noteworthy, all the cells utilized in this study are commercially available or from biorepositories. Consequently, they are excluded from the Human Research Act (HRA) and do not require ethical approval. Concerning the donors, no disease was diagnosed at the time of the biopsy. Therefore, they are all considered healthy. The iPSCs were maintained under feeder-free conditions and cultured on Cellartis DEF-CS COAT-1-coated plates in the Cellartis DEF-CS culture system according to the manufacturer’s manual. The iPSCs were kept in a humidified incubator at 37 °C and 5% CO_2_ with medium replacement every day and passaged twice a week using TripLE Select, as described in the manufacturer’s protocol from Takara Bio (Kusatsu, Shiga, Japan).

### 2.3. Neural Induction of Human iPSCs and Neuronal Differentiation

Neural induction of young and aged iPSCs to neural progenitor cells (NPCs) was conducted using the PSC Neural Induction protocol from Gibco according to the manufacturer’s instructions. Briefly, iPSCs were plated into Cellartis DEF-CS COAT-1-coated 6-well cell culture plates at a density of 2.6 × 10^5^ cells per well in the Cellartis DEF-CS culture system. After 6 h (day 0), the medium was switched to neural induction medium (NIM), containing Neurobasal Medium and 2% Neural Induction Supplement. The NIM was changed every other day from day 0 to day 4 of neural induction, followed by an everyday NIM replacement after day 4, as cells reached confluency. On day 7, primitive NPCs were dissociated with StemPro Accutase and plated into Geltrex-coated 6-well cell culture plates at a density of 4.8–9.6 × 10^5^ cells per well in neural expansion medium (NEM), which consisted of a 1:1 ratio of Neurobasal Medium to Advanced DMEM/F-12 with 2% Neural Induction Supplement and 10 µM ROCK Inhibitor Y-27632, and the latter was used for the prevention of cell death. The NEM was changed every other day until NPCs reached confluency. Expanded NPCs were either cryopreserved in NEM with 10% DMSO or used for subsequent differentiation into neurons.

For neuronal differentiation of young and aged NPCs to neurons, the NPCs were first cultured in Geltrex-coated 6-well cell culture plates in RHB-A medium supplemented with 20 ng/mL EGF and 20 ng/mL FGF for at least two passages. Passaging was performed with StemPro Accutase and 10 µM ROCK Inhibitor Y-27632 was added as overnight treatment at the time of splitting for NPCs with a passage lower than five. For differentiation into neurons, StemPro Accutase-dissociated NPCs were either plated into Geltrex-coated 6-well cell culture plates at a density of 7.6 × 10^4^ cells per well or into Geltrex-coated 10 cm^2^ cell culture dishes at a density of 4.5 × 10^5^ cells per dish in RHB-BASAL medium supplemented with 0.5% NDiff N2, 1% B-27 Supplement, and 10 ng/mL FGF. Again, 10 µM ROCK Inhibitor Y-27632 was added as an overnight treatment to prevent cell death. Half of the culture medium was replaced with fresh medium every other day for 6 days. On day 7, the differentiation medium was switched to a 1:1 ratio of RHB-BASAL medium to Neurobasal Medium minus phenol red supplemented with 0.25% NDiff N2, 1% B-27 Supplement, 10 ng/mL FGF, and 0.5% GlutaMAX. Half of the culture medium was changed on alternate days until day 13. On day 14, the differentiation medium was switched to Neurobasal Medium minus phenol red supplemented with 2% B-27 Supplement, 1% GlutaMAX, and 20 ng/mL BDNF. The cells were cultured for an additional 14 days with half of the medium replaced every other day.

### 2.4. Post-Differentiation Replating of iPSC-Derived Neurons for Experiments

To perform the different experiments, young and aged differentiated neurons were replated into the corresponding assay plates, with the aim of increasing their viability and improving the reproducibility in a small culture volume. Briefly, fully differentiated neurons were dissociated with StemPro Accutase containing 10 µM ROCK Inhibitor Y-27632 (for 45 min at 37 °C) and replated either into Geltrex-coated 96-well cell culture plates at a density of 6.0 × 10^5^ cells per well or into Geltrex-coated Seahorse XFp Cell Culture Miniplate at a density of 3.0 × 10^5^ cells per well in Neurobasal Medium minus phenol red supplemented with 2% B-27 Supplement, 1% GlutaMAX, and 20 ng/mL BDNF. To prevent cell death, 10 µM ROCK Inhibitor Y-27632 was added as an overnight treatment. Half of the culture medium was replaced with fresh medium every other day for 7 days before starting the treatment. To confirm that the iPSCs were correctly differentiated into neurons, cells were stained with the neuronal marker β-tubulin.

### 2.5. Treatment Paradigm

To investigate a potential amelioration in mitochondrial function, the impact of SPD on young and aged iPSC-derived neurons was examined. The stock solution of SPD was prepared in distilled water with a concentration of 10 mM. After 7 days of replating, young and aged neurons were treated with different concentrations of spermidine for 48 h. Briefly, the medium of the neurons was replaced with either final concentrations of SPD (0.1 µM, 0.5 µM, 1 µM, 2 µM) in Neurobasal Medium minus phenol red supplemented with 2% B-27 Supplement, 1% GlutaMAX, and 20 ng/mL BDNF or with medium alone for the untreated control condition (control). After 24 h, half of the medium was exchanged with fresh medium containing either the different concentrations of SPD or the medium alone. All experiments were conducted on the following day.

### 2.6. CellTracker Blue Dye Loading

The stock solution of CellTracker Blue was prepared in DMSO with a concentration of 10 mM. The iPSC-derived neurons were incubated in the dark with the dye at a final concentration of 5 µM for 30 min at 37 °C. After replacing the loading medium with HBSS, the fluorescence signal was detected at 353 nm (excitation)/466 nm (emission) using the Cytation 3 Cell Imaging Multi-mode Plate Reader from BioTek (Winooski, VT, USA). For the ATP assays, the dye loading was conducted before the ATP measurement, whereas for all other experiments, it was performed thereafter. The quantified CellTracker Blue signal was used to normalize the obtained data from the different experiments.

### 2.7. ATP Levels

The total ATP content was quantified using the ATPlite 1step Luminescence Assay according to the instructions of the manufacturer. The assay measures the production of light, which is formed through the reaction of ATP with luciferin, catalyzed by the enzyme luciferase. After 48 h of SPD treatment, the wells for the ATP standard curve were prepared, followed by the addition of ATP substrate solution into every well. After incubation in the dark for 2 min at room temperature under agitation, the luminescence was measured using the Cytation 3 Cell Imaging Multi-mode Plate Reader (BioTek). The emitted light was linearly correlated to the ATP concentration and the data were normalized on the quantified CellTracker signal.

### 2.8. Determination of MMP

Changes in MMP were assessed using the potentiometric fluorescent dye TMRM. After 48 h of SPD treatment, the iPSC-derived neurons were loaded in the dark with the dye at a final concentration of 0.4 µM for 30 min at room temperature under agitation. After washing twice with HBSS, the fluorescence signal was detected at 531 nm (excitation)/595 nm (emission) using the Cytation 3 Cell Imaging Multi-mode Plate Reader (BioTek), and the data were normalized on the quantified CellTracker signal.

### 2.9. Detection of ROS Levels

The levels of total mitochondrial ROS and the specific levels of O_2_^•−^ were determined using the fluorescent dye DHR and the Red Mitochondrial Superoxide Anion Indicator (MitoSOX), respectively. After 48 h of SPD treatment, the iPSC-derived neurons were loaded in the dark with a final concentration of 10 µM DHR for 15 min at room temperature or with 5 µM MitoSOX for 2 h at 37 °C under agitation. Afterward, the plates were washed twice with HBSS before measuring. During incubation, DHR and MitoSOX form fluorescent products, which were detected at 485 nm (excitation)/535 nm (emission) and 531 nm (excitation)/595 nm (emission), respectively, using the Cytation 3 Cell Imaging Multi-mode Plate Reader (BioTek). The fluorescence intensities were proportional to the total mitochondrial ROS levels and the specific O_2_^•−^ levels in mitochondria. The data were normalized on the quantified CellTracker signal.

### 2.10. Profiling Mitochondrial Respiration

Crucial parameters linked to mitochondrial respiration were evaluated using the Seahorse XF HS Mini Analyzer from Agilent Technologies (Santa Clara, CA, USA), allowing the simultaneous real-time measurement of the oxygen consumption rate (OCR) and the extracellular acidification rate (ECAR). After 48 h of SPD treatment, the XF Mito Stress Test protocol was conducted as directed by the manufacturer. For the measurement, the assay medium consisted of the Seahorse XF DMEM medium, pH 7.4 (Agilent Technologies) supplemented with 18 mM glucose, 4 mM pyruvate, and 2 mM L-glutamine. The OCR and ECAR were recorded simultaneously first under basal conditions, followed by the sequential injection of oligomycin (1.5 µM), carbonyl cyanide-p-trifluoromethoxyphenylhydrazone (FCCP, 1 µM), and a combination of antimycin A (0.5 µM) and rotenone (1 µM). The obtained data were analyzed on the Agilent Seahorse Analytics website, which automatically calculated the bioenergetic parameters, including basal respiration, maximal respiration, spare respiratory capacity, non-mitochondrial oxygen consumption, and ATP-production coupled respiration. The data were normalized on the quantified CellTracker signal.

### 2.11. Statistical Analysis

Data are presented as the mean ± SEM. Statistical analyses and data presentation were performed using Graph Pad Prism 9 (version 9.3.1). Values were normalized to the corresponding untreated control condition or the young untreated cells (=100%). For statistical comparisons of two groups, Student’s unpaired t-test was used. For statistical comparisons of more than two groups, one-way ANOVA was used, followed by Dunnett’s multiple comparison tests versus the control group. *p*-values < 0.05 were considered statistically significant. Statistical parameters can be found in the figure legends.

## 3. Results

### 3.1. Aged iPSC-Derived Neurons Display Mitochondrial Bioenergetic Deficits and Augmented Mitochondrial ROS Levels

The progressive decline of mitochondrial function indicates a well-described hallmark of neuronal aging. In particular, mitochondrial impairments ranging from decreased bioenergetics to increased oxidative stress have been reported to substantially contribute to the aging process [25]. Considering the controversial notion that reprogramming aged somatic cells to iPSCs resets metabolic and stress mechanisms in mitochondria to a more youthful state [26], we first aimed to validate our cellular aging model regarding the presence of a mitochondrial aging signature. For this reason, we differentiated human iPSCs into neurons over 5 weeks from four young and four aged healthy donors for the subsequent evaluation of crucial parameters related to mitochondrial function. To investigate whether the efficiency of mitochondrial respiration and cellular bioenergetics is compromised in aged neurons, we performed the Seahorse XF Cell Mito Stress Test using the Seahorse XF HS Mini Analyzer. Concretely, we monitored in real time the OCR (Figure 1a), an indicator of mitochondrial OXPHOS, and the ECAR (Figure 1b), an indicator of glycolysis, simultaneously. Under basal conditions, we found a significant reduction in the OCR (Figure 1c) and an even more profound decline in the ECAR (Figure 1d) of aged neurons compared to young neurons. Regarding the calculated bioenergetic parameters (Figure 1e), aged neurons presented markedly lowered rates in maximal respiration, spare respiratory capacity, non-mitochondrial oxygen consumption, and ATP-production coupled respiration, indicating a profound mitochondrial metabolic impairment.

Provided that ATP can be utilized as a general marker of cellular viability and hence mitochondrial function [27], in the next step, we determined the ATP levels in neurons (Figure 1f). As hoped, compared to young neurons, aged neurons showed a slight, but significant decrease in the ATP concentration. Moreover, to drive the synthesis of ATP during OXPHOS, the proton motive force is required, which in turn depends on the proton gradient that ultimately establishes the MMP [28]. Consequently, we wanted to verify if the lowered levels were directly coupled with changes in mitochondrial activity (Figure 1g). Indeed, aged neurons demonstrated a substantial drop in the MMP when compared to young neurons.

Of note, mitochondrial activity and OXPHOS create unavoidable by-products of ROS. Especially during mitochondrial respiration, the leakage of electrons predominantly from complexes I and III of the electron transport chain (ETC) triggers the partial reduction of oxygen, forming toxic O_2_^•−^, which subsequently can be converted into other ROS [14]. Since aged neurons presented deficits in mitochondrial respiration, we thus assessed the levels of total mitochondrial ROS and the specific levels of O_2_^•−^ using the fluorescent dye DHR and MitoSOX, respectively. Strikingly, we not only detected a pronounced elevation of total mitochondrial ROS levels (Figure 1h) but also drastically increased specific O_2_^•−^ levels (Figure 1i) in aged neurons compared to young neurons.

Altogether, these data revealed that even iPSC-derived neurons from an aged donor, where the iPSCs were generated through the process of reprogramming, could still retain age-associated changes in mitochondrial function, thus displaying the aging signature in the absence of a full rejuvenation. Consequently, aged neurons exhibited impairments in mitochondrial bioenergetics, resulting in a mitochondrial respiration deficiency with concomitant ATP depletion, which was accompanied by the depolarization of the MMP and the augmented production of different mitochondrial ROS.

### 3.2. SPD Ameliorates Important Indicators of Mitochondrial Function in Young and Aged iPSC-Derived Neurons

Previous studies using different neuronal aging models described an SPD-mediated augmentation in ATP levels and MMP [20,24,25]. Therefore, we next investigated whether a 48 h treatment with various SPD concentrations (0.1 µM, 0.5 µM, 1 µM, 2 µM) could ameliorate the ATP levels and the MMP in not only aged but also young neurons. Compared to the untreated control, we detected a significant increase in the ATP production of young neurons (Figure 2a) at all examined SPD concentrations, with 2 µM SPD exerting the highest augmentation. Likewise, SPD treatment elevated the ATP concentration in aged neurons compared to the untreated control (Figure 2b) with 1 µM and 2 µM SPD, representing the only effective doses that exhibited a marked improvement. In addition, we observed a slightly more profound effect in young neurons with the 2 µM SPD treatment, in which the enhancement amounted to 19% compared to aged neurons with only 16% amelioration.

Concerning the effects of the various SPD concentrations on the MMP after the treatment for 48 h, young neurons (Figure 2c) presented a substantial augmentation in the MMP with all tested SPD concentrations compared to the untreated control. Comparatively, aged neurons (Figure 2d) treated with SPD versus the untreated control showed a pronounced elevation in the MMP for each SPD dose. Interestingly, SPD demonstrated an overall comparable efficacy in young and aged neurons. Moreover, for both groups, 2 µM SPD treatment proved to be the particularly effective dose again, resulting in an MMP increase of 30% in young and 29% in aged neurons.

Collectively, our data suggest that SPD can improve both fundamental indicators of mitochondrial function in young and aged neurons. Nonetheless, it seems that a relatively high SPD concentration is required to ameliorate the ATP levels in aged neurons, whereas all investigated SPD doses evoked a beneficial impact on the MMP. Accordingly, the SPD-induced repolarization of the MMP may thus be perceived as especially efficient in rescuing an age-associated depolarization of the MMP.

### 3.3. SPD Alleviates Mitochondrial ROS Levels in Young and Aged iPSC-Derived Neurons

Further, previous findings reported that SPD was found to suppress oxidative stress in aging mice models [18,24]. Therefore, we subsequently explored the effects of different SPD concentrations (0.1 µM, 0.5 µM, 1 µM, 2 µM) on the total mitochondrial ROS levels (DHR) and the specific levels of O_2_^•−^ (MitoSOX). After 48 h of treatment, we observed a dose-dependent reduction in total mitochondrial ROS levels in young neurons (Figure 3a) compared to the untreated control. Similarly, SPD induced a substantial attenuation in the total amount of mitochondrial ROS in aged neurons (Figure 3b) versus the untreated control. Of note, 2 µM SPD elicited the strongest decrement in young neurons with a lessening of 20%, while all tested SPD doses had the same beneficial effect in aged neurons with a decrease of 14%.

Likewise, young neurons (Figure 3c) showed an alleviation in specific O_2_^•−^ levels after a 48 h treatment compared to the untreated control. Remarkably, exposure to 2 µM SPD resulted in a maximal reduction of 26%. Regarding the impact of the various SPD doses on aged neurons (Figure 3d), compared to the untreated control, we found a significantly lowered production of specific O_2_^•−^, with 2 µM SPD representing the most effective concentration again, causing a decrease of 31%.

Overall, these results demonstrate that SPD exerts anti-oxidative properties in young and aged neurons. Consequently, SPD alleviates the overproduction of mitochondrial ROS, in particular O_2_^•−^, thereby attenuating the age-evoked increment in oxidative stress and ultimately protecting mitochondria from potential oxidative damage.

### 3.4. SPD Improves Mitochondrial Respiration in Young and Aged iPSC-Derived Neurons

Given that SPD positively modulated several aspects of mitochondrial function in our neuronal model and has been shown to enhance mitochondrial respiration in other aging models [19,23], we next evaluated its effects on the bioenergetic profile of young and aged neurons. This assessment was conducted after 48 h of treatment, focusing on SPD concentrations of 0.1 µM and 1 µM. Accordingly, we recorded in real time the OCR of young (Figure 4a) and aged (Figure 4b) neurons concurrently to the ECAR of young (Figure 4c) and aged (Figure 4d) neurons using the Seahorse XF HS Mini Analyzer. After 48 h of SPD exposure, we detected a dose-dependent increase in the basal OCR in young neurons (Figure 4e) compared to the untreated control. However, this enhancement only reached statistical significance with 1 µM SPD, resulting in a mitochondrial respiration improvement of 33%. In contrast, SPD treatment exhibited no effects on the basal ECAR of young neurons (Figure 4f). Likewise, we observed a substantial augmentation in the basal OCR of 1 µM SPD-treated aged neurons (Figure 4g) versus the untreated control with an amelioration of 27%. In parallel, we found no SPD-induced significant changes in the basal ECAR in aged neurons (Figure 4h) compared to the untreated control.

Regarding the calculated respiratory parameters after the injection of the specific inhibitors, the treatment with SPD for 48 h led in young neurons versus the untreated control (Figure 4i) to an enhancement of all metrics with 1 µM SPD, whereas 0.1 µM SPD elicited no marked effects, albeit demonstrating a tendency for an amendment, especially for the maximal respiration and the spare respiratory capacity. In particular, we found that 1 µM SPD most effectively elevated the maximal respiration by 74% and the spare respiratory capacity by 136%, while the beneficial impact on the non-mitochondrial oxygen consumption and the ATP-production coupled respiration was lower with an increase of 26% and 34%, respectively. Similarly, compared to the untreated control, all bioenergetic parameters were ameliorated in aged neurons (Figure 4j) after 48 h of treatment with 1 µM SPD, whereas 0.1 µM SPD showed no pronounced improvement, except for again presenting a tendency for an enhanced maximal respiration and spare respiratory capacity. Specifically, we observed that 1 µM SPD significantly augmented the maximal respiration, the spare respiratory capacity, the non-mitochondrial oxygen consumption, and the ATP production-coupled respiration by 55%, 88%, 26%, and 28%, respectively.

Taken together, these results indicate that SPD positively regulates mitochondrial bioenergetics in both young and aged neurons. As a result, aged neurons switched to a metabolically more active state with an ameliorated basal respiration, thereby potentially counteracting the age-dependent deficits in mitochondrial respiration. Notably, only the concentration of 1 µM SPD was able to promote these beneficial effects. Comparatively, SPD provoked a considerable bioenergetic boost in young neurons. Interestingly, SPD mediated the most striking changes in the maximal respiration and the spare respiratory capacity in both young and aged neurons, thus primarily acting on respiratory competence.

### 3.5. Correlations Between Donor Age, Bioenergetic Parameters, and SPD Treatment

To explore the relationships among donor age, mitochondrial function, and the effects of SPD treatment on iPSC-derived neurons, we conducted Pearson’s linear regression analyses for each donor’s data (Appendix A). We observed a slight but non-significant trend toward age-related decreases in ATP levels (R^2^ = 0.08856, *p* = 0.4741) and MMP (R^2^ = 0.1375, *p* = 0.3659). However, the basal OCR significantly declined with age (R^2^ = 0.5963, *p* = 0.0247), suggesting that mitochondrial respiration may be particularly susceptible to age-related deterioration.

In addition, when analyzing the relationship between ATP levels and the MMP under varying SPD concentrations, distinct patterns emerged between young and aged neurons (Appendix A). Younger neurons showed a more pronounced positive correlation between ATP and the MMP, especially at higher SPD doses (*p*-values: 0.0711, 0.1489, 0.0034, 0.0004), while aged neurons generally displayed weaker correlations (*p*-values: 0.0481, 0.0265, 0.0966, 0.8387) and no significant relationship at lower SPD concentrations. This trend may suggest a better coupling between ATP production and the MMP in young neurons, potentially reflecting more efficient mitochondrial responses to SPD treatment. These findings indicate that SPD may enhance mitochondrial function more effectively in younger neurons, whereas aged neurons exhibit a blunted response, though this remains speculative.

## 4. Discussion

Provided that the accumulation of dysfunctional mitochondria is considered to play a causative role in the aging process [14], enhancing mitochondrial function may thus pose a promising approach for delaying aging. In addition, recent evidence highlighted SPD as a compound with potential anti-aging properties [29]. Interestingly, studies conducted in humans reported a decline in SPD levels with increasing age, implying that maintaining SPD levels during aging may hence contribute to longevity [22,30,31,32]. Strikingly, one of the proposed mechanisms involved in the observed beneficial impact of SPD on decelerating aging is attributed to the counteracting of age-related mitochondrial impairments [18,19,23,24]. For this reason, in the present study, we explored SPD’s protective effects on mitochondrial function in young and aged human iPSC-derived neurons. We have found that a 48 h SPD treatment not only enhanced mitochondrial bioenergetics but also suppressed the production of mitochondrial ROS, thereby positively affecting mitochondrial performance in both age groups (Figure 5).

SPD is an abundant and naturally occurring polyamine in all living organisms, essential for cellular homeostasis and numerous biological processes, such as DNA and RNA stabilization, cell growth and proliferation, and tissue regeneration [33]. In humans, the systemic bioavailability of SPD depends on cellular biosynthesis, microbial synthesis in the intestines, and the intake of polyamine-rich food [30]. High levels of SPD are particularly contained in unprocessed plant-derived food, including wheat germ, soybeans, mushrooms, peas, hazelnuts, broccoli, and cauliflower, as well as fermented foods like soy products and certain cheeses [34]. Thus, depending on the diet, the blood SPD level in humans may highly vary [35], making an SPD-rich diet a promising strategy to support healthy aging [36]. Indeed, a prospective population-based study described an inverse relation between the nutritional intake of SPD and prolonged survival in humans [22]. Similarly, previous findings demonstrated that SPD markedly extended the lifespan of aging yeast, flies, nematodes, and human immune cells [18] and decreased mortality in mice [20,21], consequently enhancing longevity. Further, in a neuronal aging model using mouse N2a neuroblastoma cells, SPD delayed aging and increased cell viability [19].

Of great importance, mitochondria remain best known as the main production sites of cellular energy in the form of ATP via OXPHOS. During OXPHOS, electrons are transferred stepwise to the different respiratory chain complexes of the ETC, allowing them to pump protons out of the mitochondrial matrix into the intermembrane space. This in turn creates a proton gradient that establishes the MMP, which is ultimately used to drive ATP synthesis through the ATP synthase [37]. Given that high energy levels are required for neuronal metabolism, neurons rely almost entirely on the mitochondrial OXPHOS system to meet their energy demands [14]. Notably, previous studies have shown that aging negatively influences mitochondrial bioenergetics in vitro and in vivo, with an age-dependent reduction in respiratory complexes activity [38,39,40,41,42], declined mitochondrial respiration [24,43,44], and ultimately lowered ATP production [45,46]. Moreover, these observations were accompanied by the depolarization of the MMP, which was directly linked to the decrease in energy generation [47,48]. In support, we recently reported that equally to directly converted neurons (iNs) iPSC-derived neurons from the same aged source cells exhibited characteristics of a mitochondrial aging signature, including reductions in mitochondrial respiration, the cellular ATP concentration, and the MMP [49]. Complementary to this, our present investigation also confirms these findings, indicating a deficiency in mitochondrial respiration together with a loss in the MMP, finally resulting in ATP depletion.

Regarding the effects of SPD on mitochondrial bioenergetics, SPD treatment enhanced mitochondrial respiration in both young and aged neurons. Interestingly, the observed beneficial impact of SPD on OXPHOS in both age groups affected mostly the respiratory capacity. Moreover, the SPD-evoked ameliorations in the ATP concentration and the MMP in young neurons were comparable to the ones found in aged neurons. These results partially align with a recent in vivo study by Schroeder and colleagues, which demonstrated an age-associated mitochondrial respiration decline in aged mice that was partially restored with dietary SPD [23]. However, contrary to our observations, dietary SPD exhibited no positive effects on OXPHOS in young mice but had an effect in middle-aged mice, suggesting that the impact of SPD on respiration may depend on the age of the animals. Consequently, this supports the notion that SPD generally boosts respiratory competence rather than counteracting age-related deterioration in mitochondrial function. Furthermore, they reported additional mitochondrial respiration-enhancing properties of SPD in head homogenates of flies treated with the polyamine, thus providing more evidence for the SPD-induced beneficial impact on OXPHOS, even across multiple examined species. Remarkably, consistent with our study, the SPD-provoked increase in mitochondrial respiration coincided with elevated ATP levels in aged flies. Likewise, investigations in senescence-accelerated mouse prone 8 (SAMP8) mice confirmed an amelioration in ATP levels after 8 weeks of SPD treatment [24]. Comparatively, similar findings were described in an in vitro model using mouse N2a neuroblastoma cells that were treated with d-galactose (d-Gal) to establish a neuronal aging model [19]. In this study, SPD was shown to augment mitochondrial oxygen consumption, ameliorate ATP production, and improve MMP, consequently further reinforcing the protective effects of SPD on mitochondrial bioenergetics.

Apart from crucially regulating the production of cellular energy for cell survival, mitochondria are also commonly known as the main originators of ROS formation through the activity of the ETC as inevitable by-products [50]. Considering that under physiological conditions, low levels of ROS are required in many biological processes as signaling molecules, cells possess efficient antioxidative defense mechanisms to scavenge ROS into non-toxic forms [51]. However, an increased ROS generation and/or diminished antioxidant activity may trigger oxidative stress, which subsequently damages DNA, proteins, and lipids, with mitochondria as the first targets of toxicity [52]. As a consequence, this may in succession affect the ETC and aggravate ROS formation, thereby evoking a vicious cycle of oxidative stress that further causes mitochondrial damage and a decline in their function, as a result potentially accelerating aging [14,53]. Specifically, common findings in humans, animals, and reprogrammed neurons have demonstrated an age-related decrease in the antioxidative defense system [42,54,55] together with heightened ROS levels [56,57], leading to an augmentation in oxidative stress [58,59]. In the same way, we recently showed that mitochondria-produced O_2_^•−^ and the total mitochondrial ROS presented a notable rise in aged iPSC-derived neurons and iNs [49]. In accordance, our present investigation in aged iPSC-derived neurons also revealed substantially elevated levels of total mitochondrial ROS and specific O_2_^•−^.

Concerning the effects of SPD on oxidative stress in our study, we found that SPD drastically alleviated both total mitochondrial ROS and specific O_2_^•−^ levels in young and aged neurons, accordingly exhibiting potent anti-oxidative properties. Surprisingly, SPD was particularly effective in lowering the amount of specific O_2_^•−^ in aged neurons, while we detected a slightly higher efficacy in attenuating total mitochondrial ROS levels in young neurons. Nonetheless, taking into account the age-elicited excess of ROS and the greater accumulation of dysfunctional mitochondria, the smaller impact of SPD on total mitochondrial ROS in aged neurons may regardless indicate a profound protective effect. Consistent with our findings, the above-mentioned study on SAMP8 mice reported that SPD efficiently suppressed oxidative stress in the brains of aged mice [24]. Precisely, they observed a robust reduction in malondialdehyde (MDA) levels, a marker for lipid peroxidation that reflects ROS overproduction, and a concomitant amelioration of superoxide dismutase (SOD) activity, a key antioxidant enzyme responsible for the dismutation of toxic O_2_^•−^. Similarly, Eisenberg and colleagues described that SPD effectively reduced oxidative damage in aging mice [18]. In particular, they identified increased serum levels of free thiol groups following SPD supplementation, thus counteracting the typically seen age-associated decrement. Additionally, they could demonstrate an SPD-induced improvement in stress resistance provoked by heat shock or hydrogen peroxide in aging yeast.

Oxidative stress is also linked to autophagy regulation, as ROS may induce autophagy to mitigate oxidative damage and support cytoprotection [60,61]. Likewise, the loss of MMP serves as the primary initiator to target mitochondria for selective autophagic elimination via mitophagy [62]. In general, autophagy constitutes a fundamental cellular process to recycle and clear redundant or dysfunctional cellular components through lysosomal-dependent degradation, ultimately promoting cellular survival, development, and homeostasis [63]. Consequently, mitophagy refers to a subtype of macroautophagy that selectively removes damaged or energy-deficient mitochondria, hence serving as an important mitochondrial quality control mechanism to maintain a healthy mitochondrial population and contribute to cell viability [64]. Strikingly, it has been shown that during aging, the autophagic activity for both mechanisms progressively declines [1,65], resulting in the accumulation of impaired and respiratory-defective mitochondria, thus compromising neuronal function and ultimately accelerating aging [66].

The majority of the aforementioned studies proposed an SPD-mediated boost in autophagy as the dominant mechanism of action, conceivably being responsible for the observed life-span prolongation and the delay in aging in multiple model organisms [18,19,23,24,67,68,69]. However, despite providing invaluable insights on a molecular level, the exact processes that account for the beneficial effects of SPD on autophagy and mitophagy remain to be unraveled further. Nevertheless, one of the suggested pathways involved an SPD-induced inhibition of histone acetyltransferase (HAT) activity, leading to histone H3 deacetylation, which in turn influences the epigenetic regulation of gene transcription, thereby enabling the upregulation of various autophagy-relevant transcripts, eventually provoking autophagy [18]. Another key mechanism comprised an SPD-elicited elevation of AMP-activated protein kinase (AMPK) phosphorylation levels and a concurrent positive regulation of autophagy proteins, such as LC3, Beclin 1, and p62, cooperatively stimulating autophagy again [19,24]. Moreover, SPD was found to mediate mitochondrial protection via the modulation of the PINK1 and Parkin-dependent quality control pathway, which is associated with mitophagy [23]. Interestingly, the latter study also reported that dietary SPD supplementation improved cognitive function, including spatial and temporal memory in aged mice and flies, and was likewise correlated with a lower risk for cognitive impairment and decline in humans. Comparatively, chronic SPD feeding in SAMP8 mice ameliorated memory retention loss and thus cognitive dysfunction [24]. Given that aging is accompanied by a gradual diminution in brain function [70], often manifesting in hippocampus-connected cognitive deterioration [71], age-related alterations in memory formation have been linked to mitochondrial dysfunction [71], implying that cognitive impairments may be associated with malfunctioning mitochondria during aging. Accordingly, the above-mentioned studies proposed that the SPD-mediated improvements in memory performance and cognition require functional mitochondria and autophagy [23,24], and, as previously noted, both were demonstrated to be positively regulated by spermidine supplementation. Intriguingly, a recent report of our group revealed that SPD rescued disease-associated tau-provoked bioenergetic and mitophagy deficits, hence further underlining the protective effects of SPD on mitochondria even under pathological conditions [72].

## 5. Conclusions

In summary, the present study provides more insight into how SPD boosts mitochondrial function, specifically enhancing neuronal bioenergetics and suppressing mitochondrial ROS. Considering that we not only observed a beneficial impact on aged neurons but also young ones, our findings suggest that nutritional SPD supplementation might represent an attractive therapeutic approach to prevent and delay age-related mitochondrial impairments, consequently conceivably promoting healthy longevity. In this regard, a recent study proved the safety and tolerability of enhanced SPD supplementation in elderly humans using an SPD-rich plant extract [73]. In addition, although dietary SPD supplementation presents a simple and inexpensive possibility, SPD also became available as a nutraceutical [74].

Nevertheless, several limitations should be considered. First, expanding the number of donors in future studies could help strengthen our findings, particularly in assessing the significance of age and treatment-related correlations. Moreover, while we did not specifically examine sex differences in this study, the sample size in both the young and aged groups was limited, with four males in the young group and three females and one male in the aged group. Although previous research has not identified sex-related differences in mitochondrial function [75,76,77], further investigations into potential sex-specific effects may be necessary. Lastly, our focus was solely on mitochondrial parameters, and while an aging signature was evident, exploring other aging markers could provide a more comprehensive understanding of the effects of SPD supplementation.

## Figures and Tables

**Figure 1 antioxidants-13-01482-f001:**
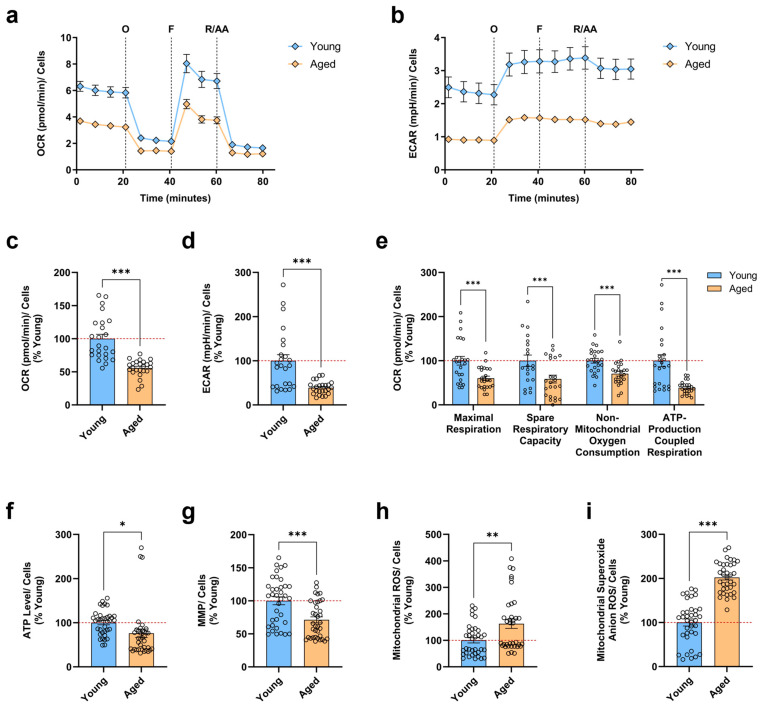
Evaluation of mitochondrial bioenergetics and ROS levels in young and aged iPSC-derived neurons. Changes over time in (**a**) OCR and (**b**) ECAR of young and aged neurons were measured simultaneously by the sequential addition of specific respiratory modulators. Quantification of (**c**) OCR and (**d**) ECAR under basal conditions in young and aged neurons from (**a**,**b**), respectively. (**e**) Respiratory parameters of young and aged neurons extracted from (**a**), specifically, maximal respiration, spare respiratory capacity, non-mitochondrial oxygen consumption, and ATP-production coupled respiration. Determination of (**f**) relative ATP levels and (**g**) MMP in young and aged neurons. Assessment of (**h**) total mitochondrial ROS levels and (**i**) O_2_^•−^ levels in young and aged neurons. Data represent the mean ± SEM of N = 3 independent experiments with n = 2 replicates per donor for (**a**–**e**) and n = 3 replicates per donor for (**f**–**i**). Values were normalized on the quantified CellTracker signal and are shown as the percentage of the young neurons for (**c**–**i**). Student’s unpaired *t*-test young versus aged, * *p* < 0.05, ** *p* < 0.01, *** *p* < 0.001. AA: antimycin A; ATP: adenosine triphosphate; ECAR: extracellular acidification rate; F: carbonyl cyanide-p-trifluoromethoxyphenylhydrazone (FCCP); MMP: mitochondrial membrane potential; O: oligomycin; OCR: oxygen consumption rate; O_2_^•−^: mitochondrial superoxide anion radical; ROS: reactive oxygen species; R: rotenone.

**Figure 2 antioxidants-13-01482-f002:**
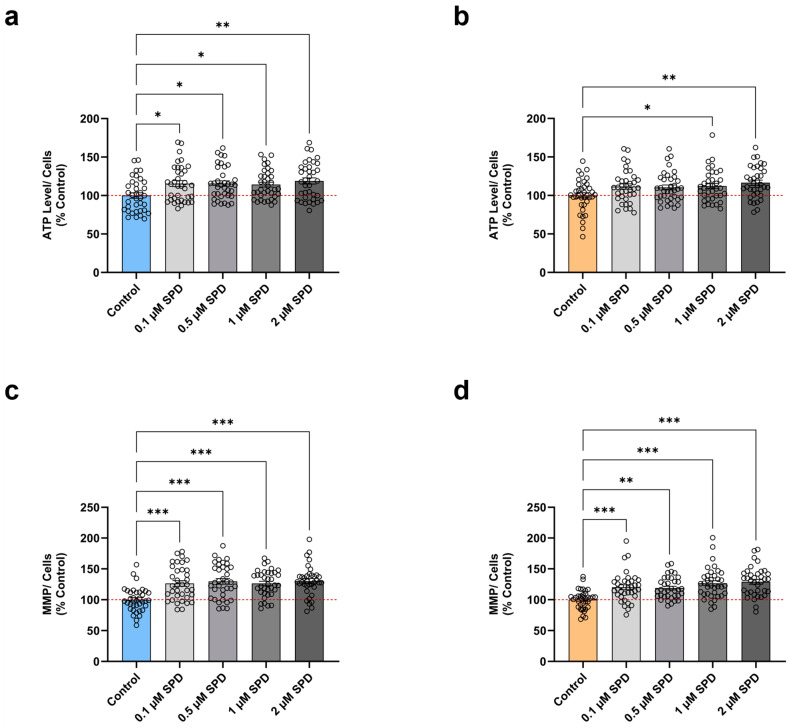
Effects of spermidine on parameters linked to mitochondrial bioenergetics in young and aged iPSC-derived neurons. Neurons were treated for 48 h with various concentrations (0.1 µM, 0.5 µM, 1 µM, 2 µM) of SPD. Quantification of relative ATP levels in (**a**) young and (**b**) aged neurons. Determination MMP in (**c**) young and (**d**) aged neurons. Data represent the mean ± SEM of N = 3 independent experiments with n = 3 replicates per donor. Values were normalized on the quantified CellTracker signal and are shown as the percentage of the control (untreated control condition). One-way ANOVA and post hoc Dunnett’s multiple comparison test versus control, * *p* < 0.05, ** *p* < 0.01, *** *p* < 0.001. ATP: adenosine triphosphate; MMP: mitochondrial membrane potential, SPD: spermidine.

**Figure 3 antioxidants-13-01482-f003:**
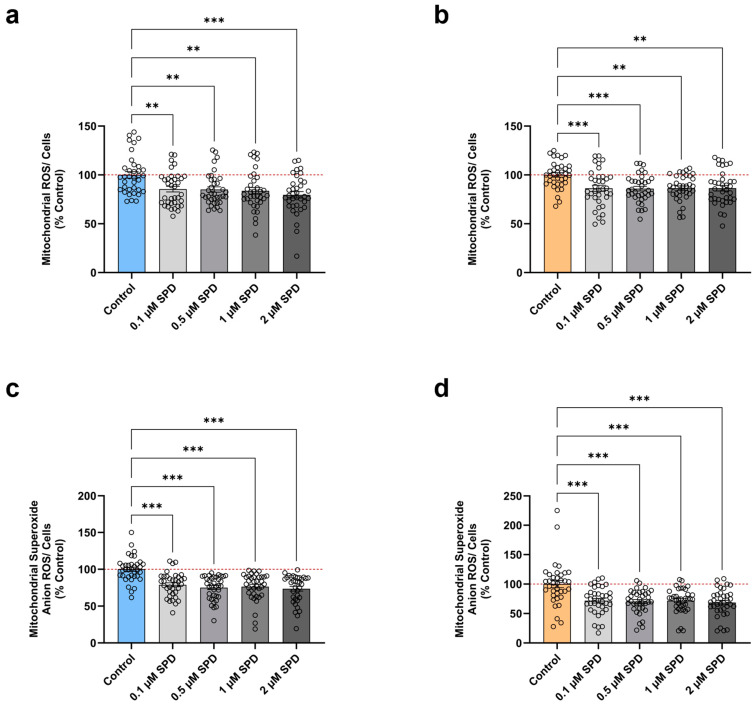
Effects of spermidine on ROS levels in young and aged iPSC-derived neurons. Neurons were treated for 48 h with various concentrations (0.1 µM, 0.5 µM, 1 µM, 2 µM) of SPD. Assessment of total mitochondrial ROS levels in (**a**) young and (**b**) aged neurons. Detection of specific O_2_^•−^ levels in (**c**) young and (**d**) aged neurons. Data represent the mean ± SEM of N = 3 independent experiments with n = 3 replicates per donor. Values were normalized on the quantified CellTracker signal and are shown as the percentage of the control (untreated control condition). One-way ANOVA and post hoc Dunnett’s multiple comparison test versus control, ** *p* < 0.01, *** *p* < 0.001. O_2_^•−^: mitochondrial superoxide anion radical; ROS: reactive oxygen species, SPD: spermidine.

**Figure 4 antioxidants-13-01482-f004:**
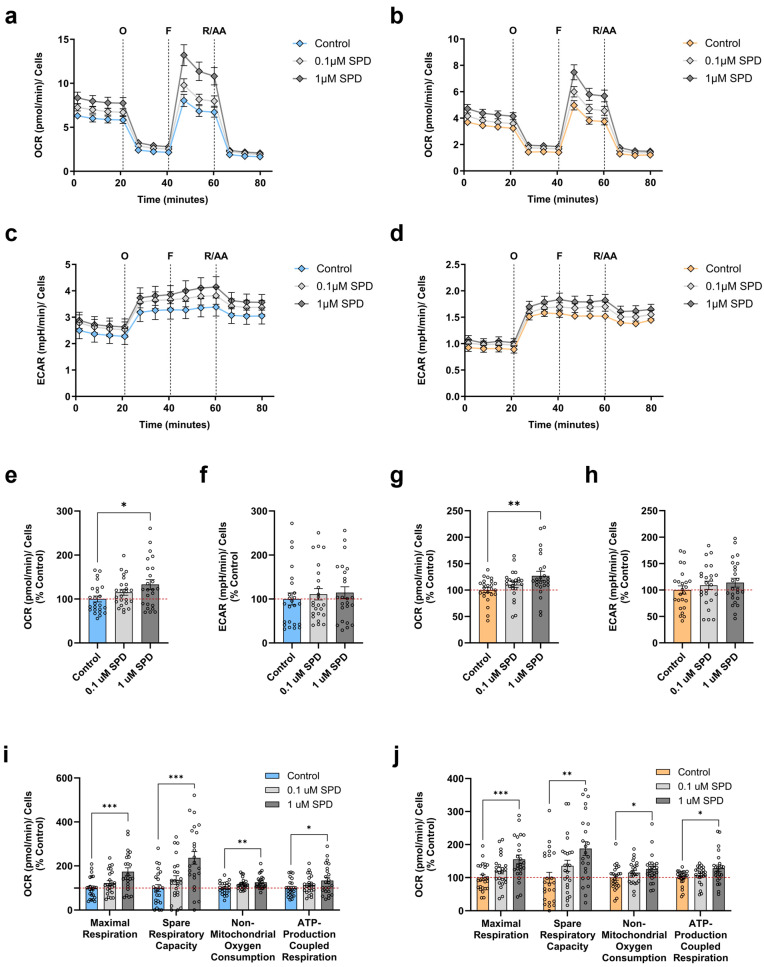
Effects of spermidine on the bioenergetic phenotype of young and aged iPSC-derived neurons. Neurons were treated for 48 h with 0.1 µM and 1 µM SPD. Changes over time in OCR of (**a**) young and (**b**) aged neurons were measured simultaneously with the ECAR of (**c**) young and (**d**) aged neurons by the sequential addition of specific respiratory modulators. Quantification of (**e**) OCR and (**f**) ECAR under basal conditions in young neurons form (**a**,**c**), respectively. Evaluation of (**g**) OCR and (**h**) ECAR under basal conditions in aged neurons form (**b**,**d**), respectively. Respiratory parameters of (**i**) young and (**j**) aged neurons extracted from (**a**,**b**), respectively, specifically, maximal respiration, spare respiratory capacity, non-mitochondrial oxygen consumption, and ATP-production coupled respiration. Data represent the mean ± SEM of N = 3 independent experiments with n = 2 replicates per donor. Values were normalized on the quantified CellTracker signal and are shown as the percentage of the control (untreated control condition). One-way ANOVA and post hoc Dunnett’s multiple comparison test versus control, * *p* < 0.05, ** *p* < 0.01, *** *p* < 0.001. AA: antimycin A; ECAR: extracellular acidification rate; F: carbonyl cyanide-p-trifluoromethoxyphenylhydrazone (FCCP); O: oligomycin; OCR: oxygen consumption rate; R: rotenone, SPD: spermidine.

**Figure 5 antioxidants-13-01482-f005:**
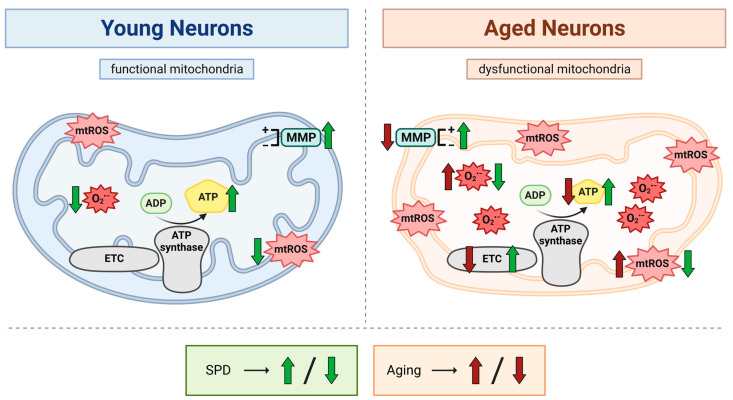
Schematic representation of the beneficial effects of spermidine on mitochondrial function in young and aged iPSC-derived neurons. Based on the findings obtained in the present study, the figure illustrates the age-related impairments in neuronal mitochondria and the SPD-mediated ameliorations in mitochondrial function for young and aged neurons. Red arrows indicate an increase or decrease in the depicted parameters for aged neurons, while green arrows denote an SPD-evoked amelioration or attenuation in the displayed parameters for young and aged neurons. ADP: adenosine diphosphate; ATP: adenosine triphosphate; ETC: electron transport chain; MMP: mitochondrial membrane potential; mtROS: mitochondrial reactive oxygen species; O_2_^•−^: mitochondrial superoxide anion radical; SPD: spermidine. Created with BioRender.com (accessed on 25 February 2023).

**Table 1 antioxidants-13-01482-t001:** Donor information of the utilized iPSC cell lines.

Category	Labeling	Line Code	Gender	Age	Origin	Source
Young	1	SF841	M	36	Human Skin Fibroblasts	Cader Laboratory
	2	Cellartis^®^ Human iPS Cell Line 12	M	24	Human Skin Fibroblasts	Takara
	3	Cellartis^®^ Human iPS Cell Line 18	M	32	Human Skin Fibroblasts	Takara
	4	Cellartis^®^ Human iPS Cell Line 22	M	32	Human Skin Fibroblasts	Takara
Aged	1	SF180	F	60	Human Skin Fibroblasts	Cader Laboratory
	2	SF854	M	72	Human Skin Fibroblasts	Cader Laboratory
	3	SF840	F	67	Human Skin Fibroblasts	Cader Laboratory
	4	SF856	F	78	Human Skin Fibroblasts	Cader Laboratory

## Data Availability

The data presented in this study are available on request from the corresponding author.

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
