# Peer review of "Spermidine Enhances Mitochondrial Bioenergetics in Young and Aged Human-Induced Pluripotent Stem Cell-Derived Neurons"

_antioxidants, 2024, doi:10.3390/antiox13121482_

Round 1
Reviewer 1 Report
In this MS, Szabo and cols. report that spermidine enhances mitochondrial function in iPSC-derived neurons of the young and adult. Mitochondrial function defined / assessed by the authors as respiratory performance, ATP production, and balanced ROS and MMP levels. This MS adds to the knowledge of the effect of Spermidine in neurons, as shown by the effects in reducing ROS and increasing ATP production.
Figures 1, 2 and 3.
1. MMP and ROS levels are assessed using a Cytation 3 Cell imaging system, which provide very accurate data based on cell images, however all results are expressed as bar diagrams. Please, show representative images for these measurements as neuronal morphology can add to the interpretation of SPD treatment.
Reviewer 2 Report
Authors reported that spermidine treatment in 19 creases adenosine triphosphate production and mitochondrial membrane potential, which is accom- 20 panied by attenuation in mitochondrial reactive oxygen species levels in both age groups. Fur- 21 furthermore, they demonstrated a spermidine-mediated amelioration in mitochondrial respiration in both 22 young and aged neurons.
I have just some additional information :
1) In the introduction section I suggest authors add more sentences to different mitochondria roles: For example, they need to discuss mitochondria dynamism since that is one of the most fundamental mechanisms in neurons. (Franco A. life 2022) And mitochondria fusion defects (Rocha AG. Science 2018). Moreover, there is also another mechanism linked to Parkin-Pink ( J Li. Frontiers in cell and developmental biology 2022)
I suggest authors to show at least one data on protein expression linked to Seahorse or mitochondria Ros production. It is possible to see SOD 1 expression levels after treatment to correlate fig. 2D-C
Reviewer 3 Report
- Brief Summary This manuscript explored a suite of mitochondrial dysfunctional differences between young and old iPSC cells, and in young and old iPSC cells before and after the application of spermidine. As there are strong indicators that mitochondria dysfunction is either a byproduct of aging or more directly responsible for cellular aging, studying the phenonemon in a human neuron model is of value. Importantly, the difference of ~30 years between the young and aged experimental groups was a strength of the manuscript.
- General concept comments
Article: Weaknesses of the manuscript was the low donor sample number, the lack of information about the donor sourced tissues, the lack of correlation analysis to individual ages of samples, the lack of sex effect testing, and the lack of images to support the quantitative analysis.
- Is the manuscript clear, relevant for the field and presented in a well-structured manner? Yes, however the Discussion could be cut for length as it is quite long and some of it is tangential to the results (lines 434-452 could be condensed, 473-498 could be condensed, lines 536-538 could be removed).
- Are the cited references mostly recent publications (within the last 5 years) and relevant? Does it include an excessive number of self-citations? Yes the references are relevant, and no it does not include an excesive number of self citations.
- Is the manuscript scientifically sound and is the experimental design appropriate to test the hypothesis? Yes, however I am concerned that the two experimental groups (young and aged neurons) are mostly pulled from distinct donor sources, making the research findings difficult to correlate to aging and not some unknown factor that could play a role in the effect the authors are seeing.
- Are the manuscript’s results reproducible based on the details given in the methods section? Yes, although the variation in the data (the major research findings) could be partially or fully due to differences in iPSC culturing techniques and protocols, donor populations. If additional information is available (iPSC culturing protocols- and how they may have differed among the two donor sources including information on the original sourced tissue, relevant health information regarding the donors), it should be provided to the extent that is appropriate and ethical.
- Are the figures/tables/images/schemes appropriate? Do they properly show the data? Are they easy to interpret and understand? Is the data interpreted appropriately and consistently throughout the manuscript? Please include details regarding the statistical analysis or data acquired from specific databases.The study would benefit from correlation of the data findings to individual samples rather than as an aggregate. This would help account for the wide age range of each experimental group. Furthermore, it would be important to understand if sex has an effect on either the age data or on the treatment data And finally, include images of the MMP analysis as well as the MitoSox analysis to accompany the quantitative data.
- Are the conclusions consistent with the evidence and arguments presented? As presented, it is difficult to definitely attribute the dysfunction seen to the age category of neurons. Age correlational analysis could possibly strengthen the arguments.
- Please evaluate the ethics statements and data availability statements to ensure they are adequate. Yes, they are.
Reviewer 4 Report
The authors should better clarify the methods used to characterize the iPSC phenotype.
Moreover, the authors should also specify the criteria used to distinguish young from old neurons and how mitochondrial function might be a parameter for this distinction.
Finally, the relationship between ATP production and aging should be further clarified.
Lines 233-236 “For this reason, we differentiated human iPSCs into neurons during 5 weeks from four young and four aged healthy donors, for the subsequent evaluation of crucial parameters related to mitochondrial function.”
Do the authors use a marker to confirm that the iPSCs were correctly differentiated into neurons?
Please, specify how the difference between young and aged neurons have been assessed apart the differeces in mitochondrial function. Do the authors use specific markers of aging?
• Lines 263-266 “Provided that ATP can be utilized as a general marker of cellular viability and hence mitochondrial function [23], in the next step we determined the ATP levels in neurons (Figure 1f). As hoped for, compared to young neurons, aged neurons showed a slight, but significant decrease in ATP concentration.”
How do the author correlate this data with aging?
Round 2
Reviewer 2 Report
No more comments to add to authors
No more comments to add
Author Response
Point 1: No more comments to add to authors.
Response 1: We thank the referee.
Reviewer 3 Report
I would like to thank the authors for including the supplemental data in order to understand the fluctuation among samples, and for adding qualifications/limitations to the overall conclusions.
*Prior to publication, please submit supplementary charts with units in the y-axes*.
If the units are added, I am satisified with the revisions and support publication in MDPI.
N/A
Author Response
Point 1: I would like to thank the authors for including the supplemental data in order to understand the fluctuation among samples, and for adding qualifications/limitations to the overall conclusions. *Prior to publication, please submit supplementary charts with units in the y-axes*. If the units are added, I am satisified with the revisions and support publication in MDPI.
Response 1: We thank the referee for pointing this out. The corresponding units were added to the charts in the Supplementary Figures S1 and S2.